

# Text-image semantic relevance identification for aspect-based multimodal sentiment analysis

Tianzhi Zhang, Gang Zhou, Jicang Lu, Zhibo Li, Hao Wu and Shuo Liu

Information Engineering University, Zhengzhou, Henan, China

## ABSTRACT

Aspect-based multimodal sentiment analysis (ABMSA) is an emerging task in the research of multimodal sentiment analysis, which aims to identify the sentiment of each aspect mentioned in multimodal sample. Although recent research on ABMSA has achieved some success, most existing models only adopt attention mechanism to interact aspect with text and image respectively and obtain sentiment output through multimodal concatenation, they often neglect to consider that some samples may not have semantic relevance between text and image. In this article, we propose a Text-Image Semantic Relevance Identification (TISRI) model for ABMSA to address the problem. Specifically, we introduce a multimodal feature relevance identification module to calculate the semantic similarity between text and image, and then construct an image gate to dynamically control the input image information. On this basis, an image auxiliary information is provided to enhance the semantic expression ability of visual feature representation to generate more intuitive image representation. Furthermore, we employ attention mechanism during multimodal feature fusion to obtain the text-aware image representation through text-image interaction to prevent irrelevant image information interfering our model. Experiments demonstrate that TISRI achieves competitive results on two ABMSA Twitter datasets, and then validate the effectiveness of our methods.

## INTRODUCTION

With the rapid development of Internet technology, online social and service platforms have gradually become an important part of people's lives (*Yu, Zhou & Gong, 2019*; *Fuji & Matsumoto, 2017*). Nowadays, instant posts and sharing personal thoughts on platforms like Twitter and Instagram are ubiquitous, the content posted by users is gradually diversified with the prevalence of social media and various service products, and people are more inclined to express their sentiment in multimodal ways such as text and image for different topics and events (*Xiao et al., 2023*). Since sentiment analysis is an effective method to extract valuable information from massive data (*Zhu et al., 2022*), multimodal sentiment analysis (MSA) task is becoming increasingly important in research communities. As an important fine-grained task in sentiment analysis, aspect-based sentiment analysis (ABSA)

Corresponding author
Gang Zhou, gzhougzhou@126.com

has attracted extensive attention from both academia and industry in the past decade for its ability to detect the sentiment polarity of the specific aspect in sample (*Zhang, Wang & Liu, 2018*; *Cao & Huang, 2023*).

Aspect-based multimodal sentiment analysis (ABMSA) is a new subtask of ABSA (*Pontiki et al., 2016*). In this article, we treat image as another modality to assist the text semantic expression, and then predict the sentiment polarity of the aspect involved in text and image. For example, given a text "Taylor posing with her Taylor Swift Award at the # BMIPopAwards" with "Taylor" as the aspect and an image of Taylor holding a trophy in her hand with a smile on her face, we can combine the context of aspect in text and the facial expression in image to predict the aspect "Taylor" as positive sentiment. Overall, ABMSA is a more refined and challenging task compared to global multimodal sentiment analysis, which can capture the sentiment polarity of text internal entities that cannot be obtained in the global tasks.

Given the importance of this task, researchers have proposed numerous methods for ABMSA. For example, *Xu, Mao & Chen (2019)* adopted attention mechanism to model interactions between aspect and text, as well as image. *Yu & Jiang (2019)*, *Yu, Jiang & Xia (2019)*, and *Wang et al. (2021)* further modeled the interactions of text-image, aspect-text, and aspect-image by employing pre-trained language and vision models. These research results demonstrate that integrating image into traditional text sentiment analysis can utilize more comprehensive sentiment information to achieve better sentiment identification effect.

Although MSA and ABSA are already popular research tasks today, ABMSA is still a relatively new research task. Employing MSA and ABSA research methods to the ABMSA task may present the following challenges: (1) Some samples in datasets have no semantic relevance between text and image. (2) Compared to text, visual feature representation extracted from image is more difficult to perform semantic expression intuitively. (3) In text-relevant images, there may also be regions that are irrelevant to the text semantics that may introduce additional interference to the model.

To address the above challenges, we propose a general multimodal architecture named Text-Image Semantic Relevance Identification (TISRI) for ABMSA. Compared with traditional ABMSA models, our model mainly achieves controlling the input of image information before multimodal feature interaction, which addresses the problems of feature complexity and interference existing in current methods, and our main contributions to TISRI are summarized as follows:

1. To improve the interaction between aspect and text as well as image, we propose a Multimodal Feature Relevance Identification (MFRI) module, which determines the relevance between text and image semantics. Since image is only used as auxiliary information for text in our research, we construct an image gate to implement dynamic input for image information to prevent text-irrelevant image interference for the model.

2. To enhance the semantic expression of visual feature representation, we construct an Image Feature Auxiliary Reconstruction (IFAR) layer that introduces adjective-noun pairs (ANPs) extracted from each image in the adopted datasets as image auxiliary information. By fine-tuning the semantic bias between image visual representation and image auxiliary

information, we can improve the image visual representation in terms of sentiment from the text level.

3. To prevent the model being influenced by irrelevant image region content, we further interact text and image representation through attention mechanism during the final multimodal feature fusion, and then obtain text-relevant image representation to achieve image feature filtering (IFF). Experimental results demonstrate that TISRI outperforms most existing advanced unimodal and multimodal methods, and achieves competitive results on two ABMSA Twitter datasets.

## RELATED WORK

Early research on sentiment analysis mainly focused on unimodal sentiment analysis of text (*Chen, 2015*; *Li & Qian, 2016*; *Shin, Lee & Choi, 2016*) and image (*You, Jin & Luo, 2017*; *Li et al., 2018*; *Wu et al., 2020*). Recently, MSA has gradually become an important focus in sentiment analysis research, and ABMSA has further developed and improved on the basis of ABSA.

### Multimodal sentiment analysis (MSA)

In recent years, MSA task has attracted widespread attention in academic community (*Cambria et al., 2017*; *Poria et al., 2020*), which aims to model text and other non-text modalities (*e.g.*, visual and auditory modalities), and generally involves two subtasks: MSA for conversation and MSA for social media. In MSA for conversation, existing methods mainly focus on adopting different deep learning models (*e.g.*, long short-term memory network (*Hochreiter & Schmidhuber, 1997*), gate recurrent unit (*Chung et al., 2014*), and Transformer (*Vaswani et al., 2017*)) to interact the information between different modalities, which have demonstrated better performance in various MSA tasks (*e.g.*, sentiment analysis (*Zadeh et al., 2017*; *Poria, Cambria & Gelbukh, 2015*; *Poria et al., 2017*; *Liang et al., 2018*), emotion analysis (*Busso et al., 2004*; *Lee et al., 2011*), and sarcasm detection (*Castro et al., 2019*; *Cai, Cai & Wan, 2019*)). In MSA for social media, it mainly includes sentiment analysis of social media image (*Chen et al., 2014b*; *You et al., 2015*; *Yang et al., 2018a*; *Yang et al., 2018b*) and multimodal sentiment analysis of text-image integration (*You et al., 2016*; *Kumar & Garg, 2019*; *Kumar et al., 2020*; *Xu, Mao & Chen, 2018*). However, the above research methods mainly focus on coarse-grained sentiment analysis (*i.e.,* identifying the global sentiment reflected by each sample) and cannot be directly employed for fine-grained ABMSA tasks.

### Aspect-based sentiment analysis (ABSA)

As an important fine-grained sentiment analysis task, ABSA has been widely researched and applied in Natural Language Processing (NLP) field over the past decade (*Cambria et al., 2017*), and its current methods can be broadly divided into two categories: discrete feature-based method and deep learning-based method. Discrete feature-based method focuses on designing multi-specific features to train learning classifiers for sentiment analysis (*Vo & Zhang, 2015*; *Pontiki et al., 2016*). Deep learning-based method mainly adopts various neural network models to encode aspects and corresponding context

information, including the method based on recursive neural network (*Dong et al., 2014*), convolutional neural network (*Xue & Li, 2018*), recurrent neural network (*Ma, Peng & Cambria, 2018*; *Chen et al., 2017*), attention mechanism (*Wang et al., 2018*; *Yang et al., 2019*; *Meškele & Frasincar, 2020*; *Zhao et al., 2021*), graph convolutional network (*Wang et al., 2020*; *Zhang & Qian, 2020*), and pre-trained BERT model that has achieved great success in the past few years (*Xu et al., 2019*; *Sun, Huang & Qiu, 2019*). However, the above research methods mainly focus on text-based unimodal information, and do not take into account the fact that relevant information from other modalities (*e.g.*, visual modality) can also contribute to sentiment analysis.

### Aspect-Based Multimodal Sentiment Analysis (ABMSA)

To conduct research on ABSA utilizing information from different modalities, researchers have developed numerous models for ABMSA in recent years by employing various effective methods in different tasks. *Xu et al. (2019)* first explored the ABMSA task and proposed a multi-interactive memory network model MIMN based on bidirectional long short-term memory (BiLSTM) network, which extracts specific aspect sentiment information from text and image, while also constructed an e-commerce comment dataset for the task. *Yu & Jiang (2019)* proposed an ABMSA model TomBERT based on the BERT (*Devlin et al., 2018*) architecture that is the basis for most extended models at present, and manually constructed two ABMSA Twitter datasets. *Yu & Jiang (2019)* proposed an ABMSA model ESAFN based on entity-sensitive attention and fusion network, which captures the relationship between aspect, text, and image information effectively. *Lu et al. (2019)* proposed a pre-trained visual language model ViLBERT that takes aspect-text pairs as input text, and the model can also be employed for the ABMSA task. *Khan & Fu (2021)* proposed an innovative model CapBERT that employs cross-modal transformation to convert the image content into text caption, and performs final sentiment analysis based on text modality only. *Wang et al. (2021)* proposed a recurrent attention network model SaliencyBERT also based on BERT, which effectively captures both intra-modal and inter-modal dynamics by designing a recurrent attention mechanism. *Zhao et al. (2022)* proposed a knowledge enhancement framework KEF based on adjective-noun pairs (ANPs) to incorporate into various models, which improves the visual attention and sentiment prediction capabilities in the ABMSA task. Although the above ABMSA methods have been validated to be effective, they often neglect to identify whether the semantics between modalities are relevant. Inspired by the above KEF framework, our model introduce ANPs as image auxiliary information, but capture the semantic relevance between modalities by calculating the similarity between text and image features before multimodal interaction, which facilitates the effective development of subsequent work.

## METHODOLOGY

In this chapter, we first formulate our task, and introduce the overall architecture of our Text-Image Semantic Relevance Identification (TISRI) model, then delve into the details of each module in TISRI.

**Task formulation:** Given a set of multimodal samples $D = (x_1, x_2, \ldots, x_d)$ as input, each sample $x_i \in D$ contains an $m$-word text $S = (w_1, w_2, \ldots, w_m)$, an associated image $I$, and an $n$-word aspect $T = (w_1, w_2, \ldots, w_n)$ that is a word subsequence of $S$. Our task is to predict the sentiment label $y \in Y$ of each given aspect, where $Y$ consists of three categories: positive, negative, and neutral.

## Overview

Figure 1 illustrates the overall architecture of TISRI, which contains the following modules: (1) Unimodal Feature Extraction Module. (2) Multimodal Feature Relevance Identification Module. (3) Aspect-Multimodal Feature Interaction Module. (4) Multimodal Feature Fusion Module.

As shown at the bottom of Fig. 1, for a given multimodal sample, we first extract word feature representations from the input text and aspect, respectively, and visual feature representation from the input image, then interact aspect representation with text and image representation to generate aspect-aware text representation and aspect-aware image representation.

Next, we obtain the semantic similarity between text and image by constructing a multimodal feature relevance identification module. The overall method is shown in Fig. 2, where the fusion representations of text and image are obtained through cross-modal interaction, and then an image gate is constructed in a specific way to dynamically control the input image information.

To enable better semantic expression of image features, we propose an image feature auxiliary reconstruction layer. As shown in Fig. 3, the image visual representation is fine-tuned by introducing ANPs extracted from each image in the adopted datasets as image auxiliary information to minimize their representation differences.

Finally, to prevent the model being influenced by irrelevant image region content, we interact aspect-aware text representation with aspect-aware image representation, and then generate the final image representation. As shown at the top of Fig. 1, we further concatenate aspect-aware text representation and final image representation, and obtain the final sentiment label through a sentiment analysis linear layer.

## Unimodal feature extraction module

In this module, we adopt two pre-trained models to extract unimodal feature representations from aspect, text and image, respectively.

## Aspect and text representation

Given an input text, we divide it into two parts: aspect $T$ and its corresponding context $C$, and replace the aspect position in $C$ with a special character "$T$". For text encoding, we employ pre-trained language model RoBERTa (*Liu et al., 2019*) as the text encoder of our model, which has been proven to achieve more competitive performance in various NLP tasks including ABSA as an enhancement to the BERT architecture (*Dai et al., 2021*). For $T$ and $C$, we follow the implementation mechanism of RoBERTa by inserting two special tokens into each input (*i.e.,* "<s>" at the beginning and "</s>" at the end), and then

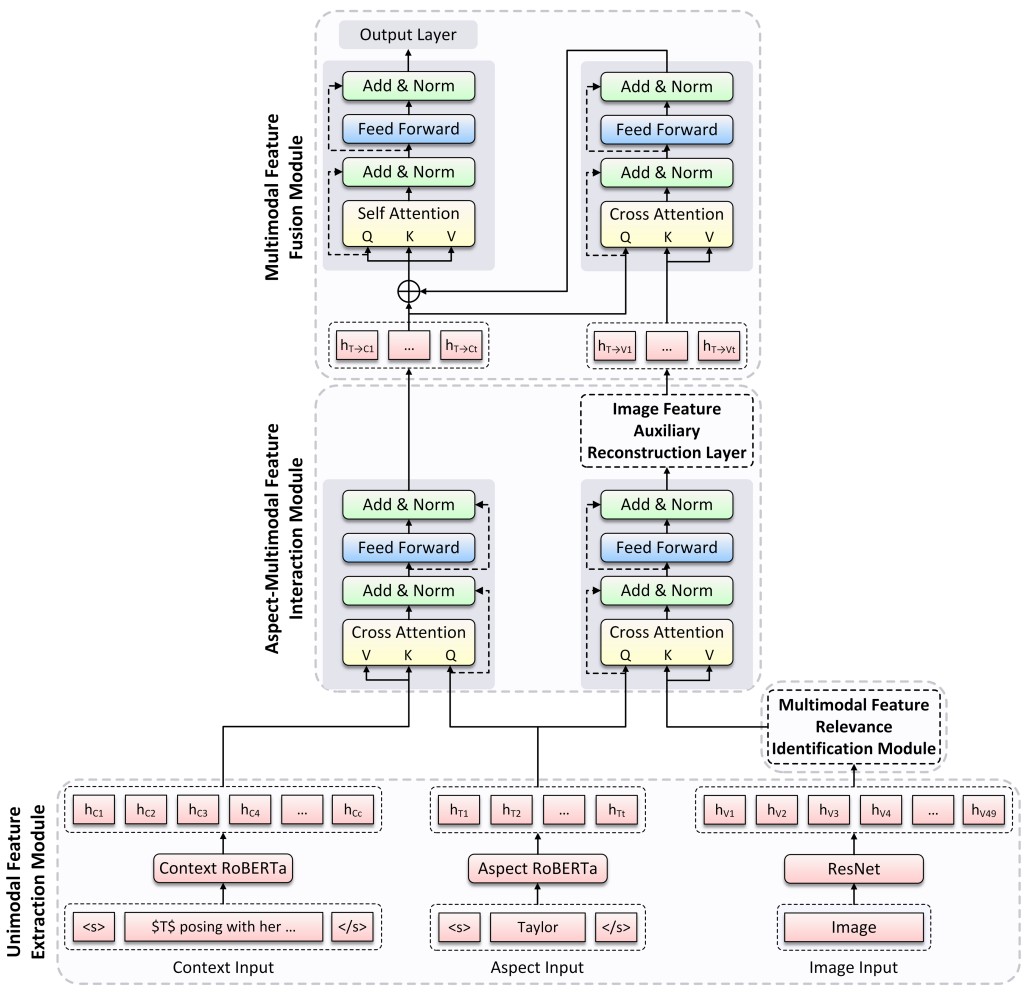

**Figure 1** **The overview of the Text-Image Semantic Relevance Identification (TISRI) model architecture.** TISRI consists of four modules: Unimodal Feature Extraction Module, Multimodal Feature Relevance Identification Module, Aspect-Multimodal Feature Interaction Module, and Multimodal Feature Fusion Module.

feeding them into text encoder to obtain the hidden representations of aspect and context:

$$H_T = \text{RoBERTa}(T) \tag{1}$$

$$H_C = \text{RoBERTa}(C) \tag{2}$$

where $H_T \in \mathbb{R}^{d \times t}$ and $H_C \in \mathbb{R}^{d \times c}$, $d$ is the hidden dimension, $t$ is the length of aspect, and $c$ is the length of context.

Next, we concatenate $C$ with $T$ as sentence $S$. For $S$, we use the token "</s>" to separate $C$ from $T$, and then obtain the hidden representation of sentence through RoBERTa as follows:

$$H_S = \text{RoBERTa}(S) \tag{3}$$

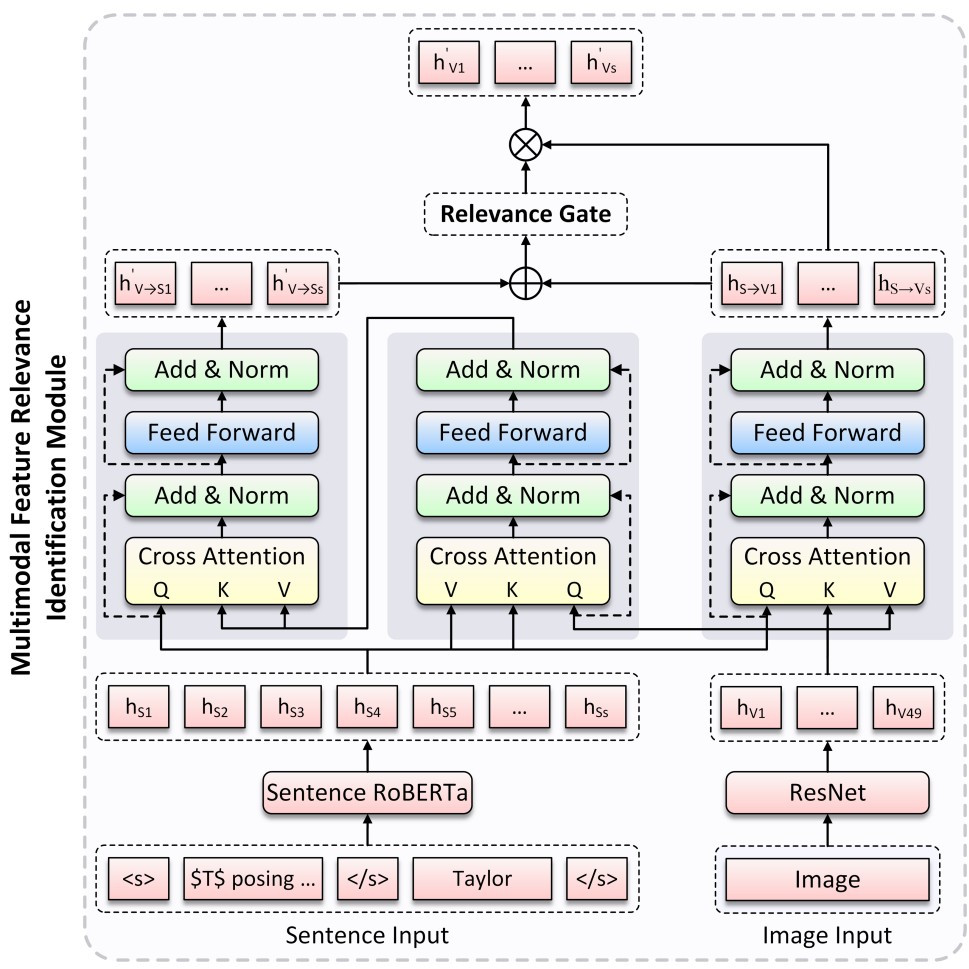

**Figure 2** The overview of the Multimodal Feature Relevance Identification (MFRI) Module architecture.

where $H_S \in \mathbb{R}^{d \times s}$, $s = c + t$ is the length of sentence. The implementations of aspect, context, and sentence encoding are shown at the bottom of Figs. 1 and 2.

## Image representation

For image encoding, we employ residual network (ResNet) (*He et al., 2016*) as the image encoder of our model. Compared to the previous VGG network (*Simonyan & Zisserman, 2014*), ResNet adopts residual connections to avoid gradient vanishing problem as the number of layers increases, which allows for deeper extraction of semantic information in image recognition tasks. Specifically, given an input image $I$, we first resize it to $I'$ with $224 \times 224$ pixels, and then take the output of the last convolutional layer in pre-trained 152-layer ResNet as the image visual representation:

$$H_I = \text{ResNet}(I') \tag{4}$$

where $H_I \in \mathbb{R}^{2,048 \times 49}$, 49 is the number of visual blocks with the same size by dividing $I'$ into $7 \times 7$, and 2,048 is the vector dimension of each visual block.

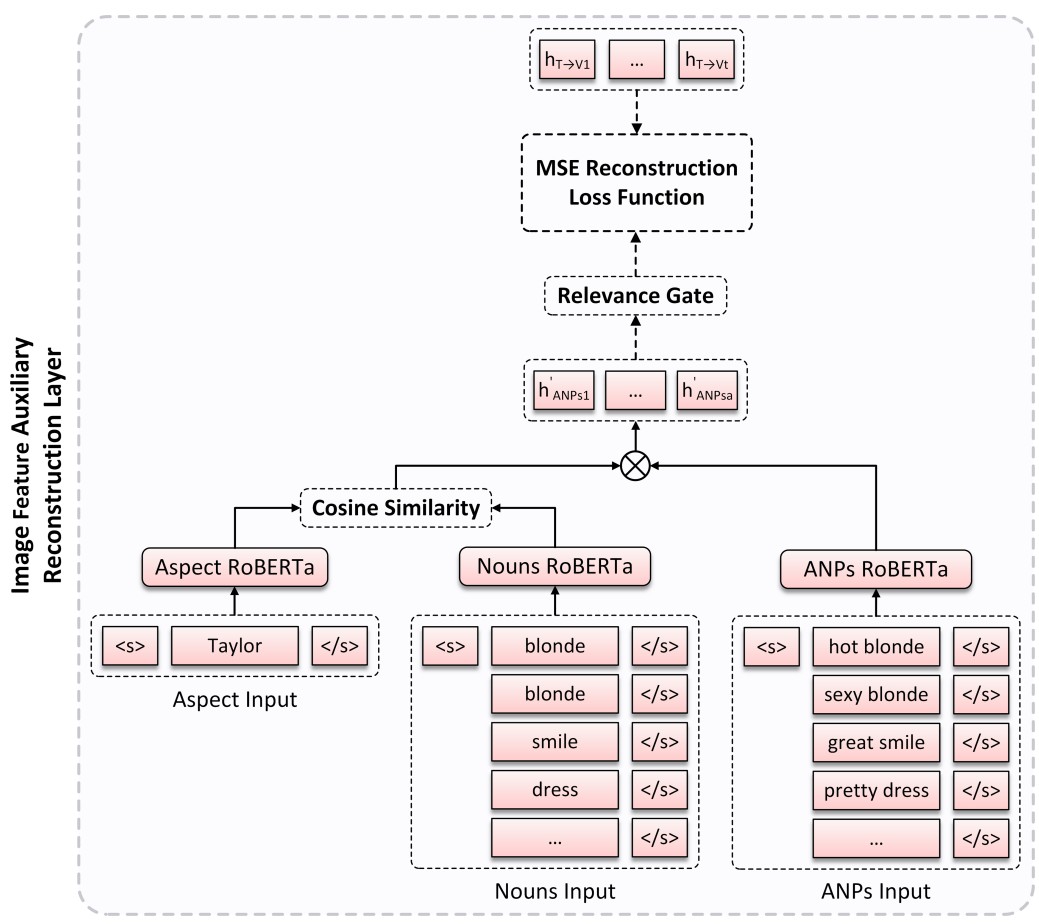

**Figure 3** **The overview of Image Feature Auxiliary Reconstruction (IFAR) layer architecture.**

Since we will subsequently conduct cross-modal interaction with text and image, it is necessary to project image representation to the same semantic space as text representation. We employ a linear transformation function for $H_I$ to obtain the projected image representation:

$$H_V = W_I^T H_I + b_I \tag{5}$$

where $W_I^T \in \mathbb{R}^{d \times 2,048}$ and $b_I \in \mathbb{R}^d$ is the learnable parameters. The implementation of image encoding is shown at the bottom of Fig. 1.

## Multimodal feature relevance identification module

For images in multimodal samples, while they can provide information beyond text, our purpose is to use image to assist in analyzing the sentiment polarity of aspect in text, and images that are irrelevant to text semantics may lead to misalignment of aspects and introduce additional interference to the model. Therefore, we propose a Multimodal Feature Relevance Identification (MFRI) module, which provides an image gate when integrating text and image, and dynamically controls the input image information based on its relevance to text semantics. The module is divided into two layers: (1) Text-Image

Cross-Modal Interaction layer. (2) Image Gate Construction layer. As shown in Fig. 2, we provide a detailed introduction to the implementation methods of these two layers in the following sections.

### Text-image cross-modal interaction layer

To better learn sentence feature representation in image, we introduce a Multi-head Cross-modal Attention (MC-ATT) mechanism (*Tsai et al., 2019*) layer, which treats image representation $H_V$ as query, and sentence representation $H_S$ as key and value, then involves two layer normalization (LN) (*Ba, Kiros & Hinton, 2016*) and a feedforward network (FFN) (*Vaswani et al., 2017*) as follows:

$$Z_{V \to S} = \text{LN}(H_V + \text{MC-ATT}(H_V, H_S)) \tag{6}$$

$$H_{V \to S} = \text{LN}(Z_{V \to S} + \text{FFN}(Z_{V \to S})) \tag{7}$$

where $H_{V \to S} \in \mathbb{R}^{d \times 49}$ is the image-aware sentence representation generated by MC-ATT layer. However, image representation is treated as query in the above MC-ATT layer, and each vector in $H_{V \to S}$ represents a visual block rather than a word representation in sentence. We expect that image-aware sentence representation can reflect on each word in sentence. Given this problem, we introduce another MC-ATT layer that treats $H_S$ as query, and $H_{V \to S}$ as key and value, then generates the final image-aware sentence representation $H'_{V \to S}$, where $H'_{V \to S} \in \mathbb{R}^{d \times s}$.

To obtain the image representation for each word in sentence, we adopt the same method as above for cross-modal interaction, treating $H_S$ as query, and $H_V$ as key and value, then generating the sentence-aware image representation $H_{S \to V}$, where $H_{S \to V} \in \mathbb{R}^{d \times s}$.

### Image gate construction layer

*Yu et al. (2020)* introduced visual gate to dynamically control the contribution of image visual features to each word in text in the multimodal named entity recognition task, and achieved effective experimental results. Inspired by this work, we construct a gate for the input image in our model, which is responsible for dynamically controlling the contribution of image information by assigning a weight in $[0, 1]$ to each image based on its relevance to corresponding sentence, (*i.e.,* preserving the higher relevance image by assigning a higher weight, and filtering the lower relevance image by assigning a lower weight) Specifically, we first concatenate $H'_{V \to S}$ and $H_{S \to V}$, and then construct the gate based on text-image relevance weight through linear transformation and nonlinear activation function:

$$g = \sigma(W^T_{S \to V}[H'_{V \to S}; H_{S \to V}] + b_{S \to V}) \tag{8}$$

where $W^T_{S \to V} \in \mathbb{R}^{d \times 2d}$ and $b_{S \to V} \in \mathbb{R}^d$ is the learnable parameters, $\sigma$ is the element-wise nonlinear activation function, which is used to control the output of $g$ in $[0, 1]$.

Based on the above image gate $g$, we can obtain the image representation that assigns relevance weight:

$$H'_V = g \cdot H_{S \to V} \tag{9}$$

## Aspect-multimodal feature interaction module

After obtaining the feature representations of aspect, context, and image, we analyze the relationships between aspect and image as well as context, respectively. Furthermore, we design an Image Feature Auxiliary Reconstruction (IFAR) layer, which serves as an auxiliary supervision for visual representation. The specific technical scheme of this module is shown at the middle part of Fig. 1, and the internal architecture of IFAR layer is shown in Fig. 3. We provide detailed implementation methods for them in the following sections.

## Aspect interaction layer

The main purpose of this layer is to obtain aspect-aware image representation and aspect-aware context representation, so we introduce MC-ATT layer to interact with aspect and image as well as context, respectively, to promote information integration between modalities. Specifically, we first conduct cross-modal feature interaction between aspect and image, treating aspect representation $H_T$ as query, and image representation $H_V$ as key and value:

$$Z_{T \to V} = \mathrm{LN}(H_T + \mathrm{MC-ATT}(H_T, H_V)) \tag{10}$$

$$H_{T \to V} = \mathrm{LN}(Z_{T \to V} + \mathrm{FFN}(Z_{T \to V})) \tag{11}$$

where $H_{T \to V} \in \mathbb{R}^{d \times t}$ is the aspect-aware image representation generated by MC-ATT layer.

Similarly, we can also obtain the aspect-aware context representation $H_{T \to C}$, where $H_{T \to C} \in \mathbb{R}^{d \times t}$.

## Image feature auxiliary reconstruction layer

To improve the effectiveness of visual feature representation, we introduce ANPs extracted from image in each sample. Since the nouns and adjectives in ANPs can reflect real content and sentiment in image to some extent, we employ them as auxiliary supervision for visual representation to obtain a more intuitive image semantic expression. Specifically, we adopt DeepSentiBank (*Chen et al., 2014a*) to generate 2,089 ANPs for each image and select the top $k$ ANPs as image auxiliary information. As the example in 'Introduction', the top five ANPs are "hot blonde", "sexy blonde", "great smile", "pretty dress" and "sexy model".

However, the extraction of image ANPs is essentially a coarse-grained extraction method, so ANPs may contain the content of image regions that are irrelevant to aspect or semantic information that is incorrectly recognized for image, and directly using these ANPs can significantly introduce additional interference to the model due to their inaccuracy. *Zhao et al. (2022)* obtained nouns relevant to aspect by calculating semantic similarity between aspect representation and ANPs noun representation in the construction of ABMSA knowledge enhancement framework, and achieved excellent alignment effect in their experiment. Inspired by this work, we concatenate the above $k$ ANPs and their corresponding nouns, respectively, and feed them into text encoder to obtain the ANPs representation $H_{ANPs}$ and the noun representation $H_N$, and then we employ cosine similarity to calculate the semantic similarity between $H_T$ and $H_N$ to achieve the aspect

alignment:

$$\alpha = \frac{H_T^T \cdot H_N}{\|H_T\| \cdot \|H_N\|} \tag{12}$$

where $\alpha$ is the similarity score between $H_T$ and $H_N$, which we use as a weight vector representing the semantic relevance of ANPs to the aspect in image. Next, we assign each individual in ANPs representation with its corresponding relevance weight to obtain the image auxiliary information representation:

$$H'_{ANPs} = \alpha \cdot H_{ANPs} \tag{13}$$

Furthermore, based on the construction of image gate $g$ in MFRI module, we also treat $g$ as image auxiliary information gate to dynamically control the contribution of ANPs to the model, and then obtain the final image auxiliary information representation:

$$H''_{ANPs} = g \cdot H'_{ANPs} \tag{14}$$

To enable visual attention to be more intuitive and accurate in representing the visual features of aspect in image, we introduce a reconstruction loss function based on mean square error (MSE) to minimize the difference between aspect-aware image representation $H_{T \to V}$ and final image auxiliary information representation $H''_{ANPs}$:

$$\mathcal{L}_R = \frac{1}{|D|} \sum_{i=1}^{|D|} (H''_{ANPs} - H_{T \to V})^2 \tag{15}$$

## Multimodal feature fusion module

In this module, we fuse aspect-aware context representation $H_{T \to C}$ and aspect-aware image representation $H_{T \to V}$ with our Image Feature Filtering (IFF) method to obtain the final aspect output representation. The implementation is shown at the top of Fig. 1. First, we introduce MC-ATT layer to implement the interaction between $H_{T \to C}$ and $H_{T \to V}$ to obtain the visual feature representation corresponding to aspect-aware context in aspect-aware image as the final aspect-aware image representation, and then filter the irrelevant region content in image:

$$Z_{T \to C \to V} = \text{LN}(H_{T \to C} + \text{MC} - \text{ATT}(H_{T \to C}, H_{T \to V})) \tag{16}$$

$$H_{T \to C \to V} = \text{LN}(Z_{T \to C \to V} + \text{FFN}(Z_{T \to C \to V})) \tag{17}$$

Next, we concatenate $H_{T \to C}$ and $H_{T \to C \to V}$, and then feed them into a multimodal self-attention layer based on Transformer for feature fusion between modalities:

$$H = \text{Transformer}(H_{T \to C}; H_{T \to C \to V}) \tag{18}$$

Finally, we feed the first token representation $H^0$ into Softmax layer to obtain the final sentiment label:

$$P(y|H) = \text{Softmax}(W^T H^0 + b) \tag{19}$$

**Table 1  The basic statistics of two Twitter datasets.** Train, Dev, and Test are short for train set, development set, and test set.

| | TWITTER-2015 | | | TWITTER-2017 | | |
|---|---|---|---|---|---|---|
| | **Train** | **Dev** | **Test** | **Train** | **Dev** | **Test** |
| Positive | 928 | 303 | 317 | 1,508 | 515 | 493 |
| Negative | 368 | 149 | 113 | 416 | 144 | 168 |
| Neutral | 1,883 | 670 | 607 | 1,638 | 517 | 573 |
| Total | 3,179 | 1,122 | 1,037 | 3,562 | 1,176 | 1,234 |
| Sentence | 2,101 | 727 | 674 | 1,746 | 577 | 587 |
| Avg Aspects | 1.348 | 1.336 | 1.354 | 1.410 | 1.439 | 1.450 |
| Words | 9,023 | 4,238 | 3,919 | 6,027 | 2,922 | 3,013 |
| Avg Length | 16.72 | 16.74 | 17.05 | 16.21 | 16.37 | 16.38 |

We adopt the cross entropy loss constructed by predicted values of aspect-based sentiment labels and their true values as the training loss function for the model sentiment analysis task:

$$\mathcal{L}_S = \frac{1}{|D|} \sum_{j=1}^{|D|} \log P(y^j | H^0) \tag{20}$$

To further optimize all parameters of our model, we train the loss function for sentiment analysis jointly with image auxiliary reconstruction, and then construct the final training loss function combining the two tasks:

$$\mathcal{L} = \mathcal{L}_S + \lambda \mathcal{L}_R \tag{21}$$

where $\lambda$ is the tradeoff hyper-parameter used to control the contribution of reconstruction loss.

## Experiment

In this chapter, we conduct extensive experiments on two ABMSA datasets to validate the effectiveness of our TISRA model.

## Experimental settings
### Datasets

We adopt two benchmark datasets of ABMSA TWITTER-2015 and TWITTER-2017 proposed by *Yu & Jiang (2019)* in our experiments, which are composed of multimodal tweets posted on Twitter in 2014–2015 and 2016–2017, each sample consists of a text, an image associated with text, a given aspect extracted from text, and the sentiment label (positive, negative, and neutral) corresponding to the aspect. For aspect selection, the two datasets manually extract specific noun phrases in sentence as aspects, and each sentence contains one or more aspects as sentiment analysis objects. For example, the sentence "David Gilmour and Roger Waters playing table football." treats the noun phrases "David Gilmour" and "Roger Waters" as aspects. The relevant information of these two datasets is shown in Table 1.

**Table 2  Hyper-parameter value setting of TISRI.**

| Hyper-Parameter | TWITTER-2015 | TWITTER-2017 |
|---|---|---|
| Batch Size | 16 | 16 |
| Training Epoch | 9 | 9 |
| $k$ Value | 5 | 5 |
| $\lambda$ Value | 0.8 | 0.8 |
| Learning Rate | 1e−5 | 1e−5 |
| Maximum Sentence Length | 128 | 128 |
| Maximum Context Length | 128 | 128 |
| Maximum Aspect Length | 32 | 32 |
| Hidden Dimension | 768 | 768 |
| Attention Head Number | 12 | 12 |

*Implementation details*

For TISRI, we adopt RoBERTa-base (*Liu et al., 2019*) as the encoder for sentence, context, and aspect in text, and ResNet-152 (*He et al., 2016*) as the image encoder. During alternating optimization process, we use AdamW as the learner to optimize parameters. The hyper-parameter values we set for our model are described in Table 2. In the experiments, we employ test set results corresponding to the optimal results on development set as the final results, and demonstrate the average results of three independent training runs for our experimental models. All the models are implemented based on Python 3.7 and PyTorch, compiled on PyCharm, and run on an NVIDIA Tesla V100 GPU.

## Compared baselines

In this section, we evaluate the performance of TISRI by comparing it to various existing methods. Specifically, we consider comparing the following unimodal and multimodal methods with our model:

● Res-Target: a baseline method for obtaining the visual feature representation of input image directly from the ResNet model.

● AE-LSTM (*Wang et al., 2016*): an attention-based LSTM model for obtaining important context relevant to aspect.

● MGAN (*Fan, Feng & Zhao, 2018*): a multi-grained attention network that fuses aspect and context at different granularity.

● BERT (*Devlin et al., 2018*): a pre-trained language model with stacked Transformer encoder layers for the interaction between aspect and text.

● RoBERTa (*Liu et al., 2019*): a further improved BERT model by adopting better training strategies and larger corpus.

● MIMN (*Xu et al. 2019*): a multi-interactive memory network for the interaction between aspect, text, and image.

● ESAFN (*Yu & Jiang, 2019*): an entity-sensitive attention and fusion network for obtaining inter-modal dynamics of aspect, text, and image.

● ViLBERT (*Lu et al., 2019*): a pre-trained visual language model that takes aspect-text pairs as input text.

• TomBERT (*Yu & Jiang 2019*): an aspect-aware ABMSA method based on multimodal BERT model architecture.
• SaliencyBERT (*Wang et al., 2021*): a recursive attention network based on multimodal BERT model architecture for ABMSA.
• CapBERT (*Khan & Fu 2021*): a method of converting image into text caption and feeding it with the input text to a pre-processed BERT model.
• KEF-TomBERT (*Zhao et al., 2022*): an extended baseline to apply a proposed knowledge enhancement framework KEF to TomBERT.
• KEF-SaliencyBERT (*Zhao et al., 2022*): an extended baseline to apply a proposed knowledge enhancement framework KEF to SaliencyBERT.
• CapRoBERTa: an extended baseline that replaces BERT with RoBERTa in CapBERT.
• KEF-TomRoBERTa: an extended baseline that replaces BERT with RoBERTa in KEF-TomBERT.

## EXPERIMENTAL RESULTS AND ANALYSIS

Table 3 demonstrates the performance of our model and each compared baseline model on TWITTER-2015 and TWITTER-2017 datasets. We adopt accuracy (Acc) and macro-F1 as evaluation metrics. As shown in the fifth and sixth last columns of Table 3, we compare our model with the latest proposed best performing KEF-TomBERT and KEF-SaliencyBERT. First, we replace the RoBERTa in TISRI with BERT, and then obtain TISRI-BERT to demonstrate the superiority of our adoption with RoBERTa as text encoder. More importantly, we also select the best performing CapBERT from original baseline model and better performing KEF-TomBERT from the two knowledge enhancement models, and replace the BERT in them with RoBERTa then obtain CapRoBERTa and KEF-TomRoBERTa to implement a more significant and fairer direct comparison of TISRI. The hyper-parameter values of CapRoBERTa and KEF-TomRoBERTa are described in Tables 4 and 5.

Based on all the experimental results in Table 3, we can conclude as follows: (1) The performance of Res-Target is lower than that of all text language models, which may be explained by the fact that image relevant to aspect mostly serve as an auxiliary role for text and do not perform well as an independent modality for sentiment prediction. (2) Most multimodal methods generally perform better than unimodal methods, which indicates that image information can complement text information to obtain more accurate sentiment prediction. (3) TomBERT, SaliencyBERT and CapBERT perform much better than other original multimodal models, and we speculate that adopting multi-head cross-modal attention and self-attention mechanism to do cross-modal interaction on aspect can obtain more robust feature representation. (4) Among all original baseline models, CapBERT achieves the best performance due to image caption, which indicates that text has a more intuitive semantic representation than image. (5) The performance of KEF-TomBERT and KEF-SaliencyBERT is better than that of all original baseline models, which indicates that the knowledge enhancement framework KEF can improve the performance of original model by introducing image adjective and noun information to some extent and has

**Table 3 Experimental results on TWITTER-2015 and TWITTER-2017 datasets using different uni-modal and multimodal methods in the ABMSA task.** The last three columns are where we focus our comparison, and the fifth and sixth last columns are the latest proposed of all demonstrated baselines. The best scores for each metric are in bold.

| Method | TWITTER-2015 | | TWITTER-2017 | |
|---|---|---|---|---|
| | Acc | Macro-F1 | Acc | Macro-F1 |
| | Image Only | | | |
| Res-Target | 59.88 | 46.48 | 58.59 | 53.98 |
| | Text Only | | | |
| AE-LSTM | 70.30 | 63.43 | 61.67 | 57.97 |
| MGAN | 71.17 | 65.51 | 64.75 | 61.46 |
| BERT | 74.15 | 68.86 | 68.15 | 65.23 |
| RoBERTa | 76.28 | 71.36 | 69.77 | 68.00 |
| | Text and Image | | | |
| MIMN | 71.84 | 65.69 | 65.88 | 62.99 |
| ESAFN | 73.38 | 67.37 | 67.83 | 65.52 |
| ViLBERT | 73.76 | 69.85 | 67.42 | 64.87 |
| TomBERT | 77.15 | 71.75 | 70.34 | 68.03 |
| SaliencyBERT | 77.03 | 72.36 | 69.69 | 67.19 |
| CapBERT | 78.01 | 73.25 | 69.77 | 68.42 |
| KEF-TomBERT | 78.68 | 73.75 | 72.12 | 69.96 |
| KEF-SaliencyBERT | 78.15 | 73.54 | 71.88 | 68.96 |
| TISRI-BERT | 78.21 | 74.10 | 72.12 | 70.63 |
| CapRoBERTa | 77.82 | 73.38 | 71.07 | 68.57 |
| KEF-TomRoBERTa | **78.75** | 73.94 | 72.18 | 70.21 |
| TISRI (Ours) | 78.50 | **74.42** | **72.53** | **71.40** |

**Table 4 Hyper-parameter values of CapRoBERTa.**

| Hyper-Parameter | TWITTER-2015 | TWITTER-2017 |
|---|---|---|
| Batch Size | 16 | 16 |
| Training Epoch | 6 | 6 |
| Learning Rate | 5e−5 | 5e−5 |
| Tokenizer Maximum Length | 80 | 80 |
| Hidden Dimension | 768 | 768 |
| Attention Head Number | 12 | 12 |

excellent compatibility effect. (6) Since RoBERTa is more powerful than BERT, intuitively the overall performance of CapRoBERTa and TISRI is generally better than that of CapBERT and TISRI-BERT on the above evaluation metrics. (7) Compared to the best performing KEF-TomRoBERTa, TISRI achieves competitive results on the two datasets, which has about 0.5% higher macro-F1 on TWITTER-2015 dataset, and about 0.4% and 1.2% higher accuracy and macro-F1 on TWITTER-2017 dataset, respectively. The results indicate that our model can improve the overall performance by dynamically controlling the input image information while adopting ANPs as image auxiliary information.

**Table 5  Hyper-parameter values of KEF-TomRoBERTa.**

| Hyper-Parameter | TWITTER-2015 | TWITTER-2017 |
|---|---|---|
| Batch Size | 16 | 16 |
| Training Epoch | 8 | 5 |
| Learning Rate | 2e−5 | 2e−5 |
| Maximum Sentence Length | 64 | 64 |
| Maximum Aspect Length | 32 | 32 |
| Hidden Dimension | 768 | 768 |
| Attention Head Number | 12 | 12 |

**Table 6  Ablation study of TISRI. IFF, IFAR, and MFRI are our main designed units.** The best scores for each metric are in bold.

| Method | TWITTER-2015 | | TWITTER-2017 | |
|---|---|---|---|---|
| | Acc | Macro-F1 | Acc | Macro-F1 |
| TISRI | **78.50** | **74.42** | **72.53** | **71.40** |
| TISRI w/o IFF | 76.57 | 72.22 | 71.07 | 69.74 |
| TISRI w/o IFAR | 76.37 | 72.54 | 70.02 | 68.24 |
| TISRI w/o MFRI | 77.82 | 73.85 | 70.75 | 68.76 |
| TISRI w/o IFF & IFAR & MFRI | 76.28 | 71.79 | 68.07 | 66.86 |

For the slightly lower accuracy of our model on TWITTER-2015 dataset compared to KEF-TomRoBERTa, we speculate the possible reason is that KEF-TomRoBERTa applies adjectives in the obtained ANPs directly to aspect-aware image representation, while the overall text-image relevance weights on TWITTER-2015 dataset are relatively higher than those on TWITTER-2017 dataset, which is also validated in the TISRI w/o MFRI part of ablation study in 'Value of $k$', so the direct use of adjectives can express the sentiment in image more intuitively to some extent in this case. However, for the condition where text-image relevance has a low weight, KEF-TomRoBERTa may introduce additional interference to the model by directly using irrelevant adjectives, so we speculate that TISRI performs better on TWITTER-2017 dataset for this reason.

## Ablation study

To further investigate the impact of individual unit in TISRI on model performance, we perform ablation analysis on TWITTER-2015 and TWITTER-2017 datasets for the main designed units in our model: (1) Image Feature Filtering (IFF) method. (2) Image Feature Auxiliary Reconstruction (IFAR) layer. (3) Multimodal Feature Relevance Identification (MFRI) module. We first remove the above three units respectively, and then remove these units at the same time leaving only the bone framework, so we can have a clearer and more comprehensive understanding of the contribution of individual unit to the model performance improvement. The experimental results are shown in Table 6, where w/o represents the removal of corresponding unit.

First, we can learn that removing IFF unit decreases accuracy by about 1.9% and 1.5% on the two datasets, respectively, which validates that retaining useful information in image

and implementing filtering on text-irrelevant image region content helps reduce the impact of interference on model performance. Next, removing IFAR unit decreases accuracy by about 2.1% and 2.5% on the two datasets, respectively, which proves that the unit has a large contribution to model performance improvement and validates that adopting ANPs as image auxiliary information can be more intuitive for semantic expression of visual feature representation. Then, removing the MFCR unit decreases accuracy by about 0.7% and 1.8% on the two datasets, respectively, which validates that assigning image to an inter-modal relevance weight can prevent additional interference to the model from text-irrelevant images. We can also learn that there are more images with higher text-image relevance weights in TWITTER-2015 dataset than in TWITTER-2017 dataset, which validates the reason we inferred in 'Values of epoch and batch size'. Finally, we remove all above units and observe that accuracy decreases by about 2.2% and 4.5% on the two datasets, respectively, which validates the effectiveness of our designed units in the model and also validates that these units contribute to model performance improvement to some extent from another perspective.

## Parameter analysis

In this section, we provide a detailed introduction and analysis of the process of evaluating optimal hyper-parameters. All the above experiments are set based on optimized model hyper-parameters.

### Values of epoch and batch size

To analyze the impact of different epoch and batch size on model performance, we determine the final values of epoch and batch size through experiments in this subsection. Figures 4 and 5 demonstrate the model performance of different epoch and batch size values on the two datasets, respectively, and we can draw the following inferences. First, we experiment with the value of epoch. We find that as the value of epoch increases, the model performance shows an upward trend and then gradually stabilizes. The model performance is optimal when epoch equals 8, and then accuracy and macro-F1 start to gradually decrease when epoch equals 9. Thus, we set the value of epoch to 9 in our experiments. Accuracy and macro-F1 corresponding to the epoch setting of TISRI on TWITTER-2015 and TWITTER-2017 datasets are shown in Figs. 4A and 4B.

Then, we analyze the value of batch size using 8, 16 and 32, respectively, and experimental results on the two datasets are shown in Table 7. We can clearly find that the model achieves the best performance on both datasets when batch size equals 16. The possible reasons are speculated as follows: When batch size equals 8, it is small for the number of samples in the two datasets, and the model training is not only time-consuming but also difficult to converge, which leads to the underfitting of model. In a certain range, the increase of batch size is conducive to the stability of model convergence. However, when batch size equals 32, the model may fall into local minimum because it is too large, which leads to the deterioration of model generalization performance. Thus, we set the value of batch size to 16 in our experiments.

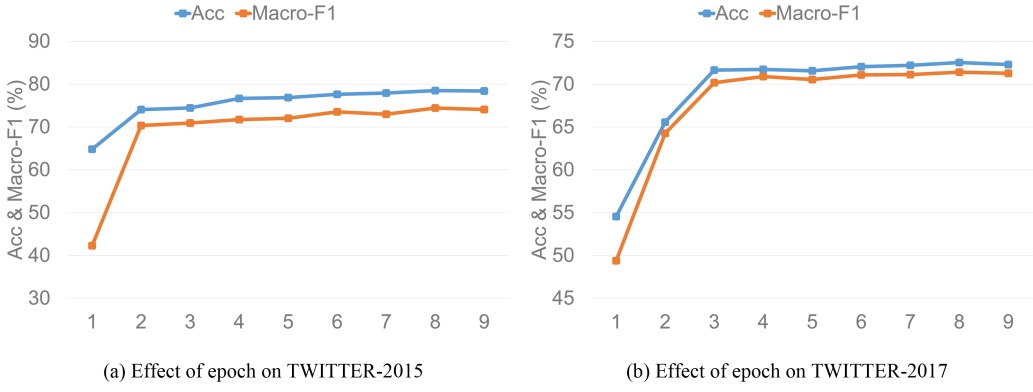

(a) Effect of epoch on TWITTER-2015      (b) Effect of epoch on TWITTER-2017

**Figure 4**   **Effect of epoch on model accuracy and macro-F1.**

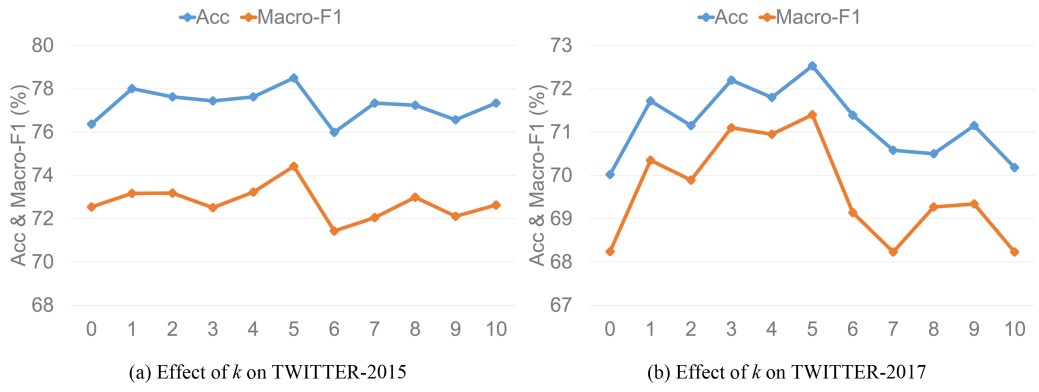

(a) Effect of $k$ on TWITTER-2015      (b) Effect of $k$ on TWITTER-2017

**Figure 5**   **Effect of k on model accuracy and macro-F1.**

**Table 7**   **Effect of batch size on model accuracy and macro-F1.** The best scores for each metric are in bold.

| Batch size | TWITTER-2015 | | TWITTER-2017 | |
|---|---|---|---|---|
| | Acc | Macro-F1 | Acc | Macro-F1 |
| 8 | 77.05 | 72.89 | 70.99 | 69.89 |
| 16 | **78.50** | **74.42** | **72.53** | **71.40** |
| 32 | 76.37 | 71.80 | 69.85 | 68.11 |

## Value of *k*

To explore the impact of ANPs on model performance, we extract the top $k$ ANPs for each image where $k$ is set as each integer in $[1, 10]$, and take values for them respectively to experiment. Figures 5A and 5B demonstrate the model performance of $k$ value on the two datasets, respectively, and we can draw the following inferences. First, the model performance is poor without ANPs as image auxiliary information, which indicates that combining ANPs can improve the performance of TISRI. Second, the model performance

shows a fluctuating upward trend as the number of ANPs increases and reaches the best state when $k$ equals 5. However, the model performance no longer improves but shows a trend of decline when $k$ is greater than 5. The possible reason is speculated as follows: The number of aspects involved in each text in the two datasets may not exceed 5, and when $k$ is greater than it, image auxiliary information may introduce additional interference to the model. Therefore, we set the value of $k$ to 5 in our experiments.

## Value of $\lambda$

To investigate the effect of trade-off hyper-parameter $\lambda$ that controls the auxiliary reconstruction loss contribution of IFAR layer on model performance, we set $\lambda$ to a decimal number with an interval of 0.1 in the range of [0, 1] to experiment. Figures 6A and 6B demonstrate the model performance of $\lambda$ value on the two datasets, respectively. The model performance shows a fluctuating upward trend as $\lambda$ increases, which has a more obvious effect on the TWITTER-2017 dataset. When $\lambda$ equals 0.8, the model performance reaches the best state, and then decreases gradually as $\lambda$ increases. We speculate that the possible reason is that ANPs as image auxiliary information only serve to improve the semantic expression of image visual features. When the trade-off hyper-parameter $\lambda$ exceeds a certain value, image auxiliary information plays a dominant role in image representation, but these ANPs may have semantic information of image recognition error, so the model will largely introduce additional interference when $\lambda$ is too large and produce negative effect. Thus, we set the trade-off hyper-parameter $\lambda$ to 0.8 in the error back propagation process.

## Error analysis

On the basis of the above experiments, we further perform error analysis on TISRI to deepen our understanding of model performance. The error prediction reasons for TISRI are broadly divided into the following categories: (1) The ANPs are incorrect in image semantic recognition. For example, given a text "Petition to have Jessica Lange come back for American Horror Story season 6" with "Jessica Lange" as the aspect and an image of Jessica Lange with a cigarette in her mouth, but the ANPs produce words such as "laughing" and "funny" that completely hinder correct sentiment recognition, so TISRI introduces incorrect word information into the image representation adjustments during model training, which affects the model performance to some extent. (2) The aspect in text cannot find nouns with high similarity in ANPs. For example, given a text "This morning @ SheilaGCraft hosted a brunch amp poured into our WILD Women to honor them for their leadership in 2014 !" with "WILD Women" as the aspect and an image of a group of women dining together, but "WILD Women" is actually an organization name that cannot be represented intuitively in image, which causes ANPs to recognize some aspect-irrelevant nouns, so the image is also affected by these ANPs and cannot accurately express the sentiment semantics of aspect. (3) The model cannot recognize deeper sentiment in text and image. For example, given a text "RT @ NYRangers: OFFICIAL: Martin St . Louis announces retirement from the National Hockey League . # NYR" with "Martin" as the aspect and an image of Martin waving on the field with the crowd cheering, but our model

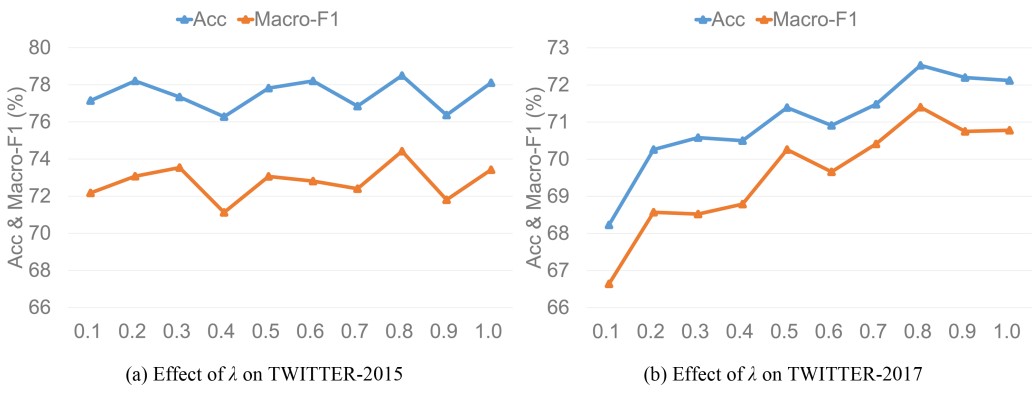

(a) Effect of λ on TWITTER-2015        (b) Effect of λ on TWITTER-2017

**Figure 6**   **Effect of λ on model accuracy and macro-F1.**

can only identify semantic features on the surface of text and image combined with ANPs, and cannot feel the deeper sentiment of Martin's unwillingness to leave the stadium, so this problem is also the difficulty and challenge for existing models to further intelligently identify sentiment in the future.

## CONCLUSIONS

In this article, we propose an aspect-based multimodal sentiment analysis (ABMSA) model TISRI, which mainly dynamically inputs image information before multimodal feature interaction to prevent text-irrelevant image interference compared with the current research in this task. First, the model calculates text-image semantic relevance and constructs an image gate that dynamically controls the input of image information. Then, it introduces adjective-noun pairs (ANPs) as image auxiliary information to enhance the semantic expression ability of image visual features. Finally, it adopts attention mechanism to interact with text and image representation to obtain filtered text-relevant image representation for the final sentiment prediction. Experimental results demonstrate that TISRI outperforms the majority of existing advanced models on the TWITTER-2015 dataset and all compared baseline models on the TWITTER-2017 dataset, and then validate the superiority of our model and the effectiveness of our methods.

However, our existing research still has the following limitations: (1) TISRI adopts ANPs as auxiliary image information, which enables visual attention to be more intuitive and accurate in representing the visual features of aspect in image, but there still exists inaccurate image descriptions in ANPs that may introduce additional interference to the image representation. (2) The two Twitter datasets adopted for the experiments are both constructed by English-based social media platforms, which are small and homogeneous, and the sentiment prediction performance of our model is unknown for datasets of different fields, languages and scales.

Considering the above limitations, we plan to expand our future research in the following directions. First, we aim to construct a more efficient method to incorporate ANPs into our model while preserving the original image features. Then, we hope to apply TISRI to more

multimodal related tasks and keep the datasets diverse to test the availability of our model in different fields and scenes, where our designed units can be easily extended to other tasks such as multimodal event extraction and multimodal named entity recognition. Moreover, with the prevalence of large language model (LLM), we intend to further explore how to effectively integrate LLM into our work and achieve more specific multiclassification tasks in subsequent research.

### Funding

This research is supported by the Science and Technology Research Program of the Department of Science and Technology of Henan Province (approval No.: 222102210081). The funders had no role in study design, data collection and analysis, decision to publish, or preparation of the manuscript.

### Grant Disclosures

The following grant information was disclosed by the authors:
Department of Science and Technology of Henan Province: 222102210081.

### Competing Interests

The authors declare there are no competing interests.

### Author Contributions

- Tianzhi Zhang conceived and designed the experiments, performed the experiments, analyzed the data, performed the computation work, prepared figures and/or tables, authored or reviewed drafts of the article, and approved the final draft.
- Gang Zhou conceived and designed the experiments, authored or reviewed drafts of the article, and approved the final draft.
- Jicang Lu conceived and designed the experiments, authored or reviewed drafts of the article, and approved the final draft.
- Zhibo Li analyzed the data, prepared figures and/or tables, and approved the final draft.
- Hao Wu performed the computation work, prepared figures and/or tables, and approved the final draft.
- Shuo Liu performed the computation work, prepared figures and/or tables, and approved the final draft.

### Data Availability

The third-party datasets from ''Adapting BERT for Target-Oriented Multimodal Sentiment Classification'' (DOI: https://doi.org/10.24963/ijcai.2019/751) are available at GitHub: https://github.com/jefferyYu/TomBERT.

The data of each tweet's associated images is available at figshare: Zhang, Tianzhi (2024). Twitter Datasets.zip. figshare. Dataset. https://doi.org/10.6084/m9.figshare.25303591.v1.

The data of tweets is available in the Supplementary File.

## Supplemental Information

Supplemental information for this article can be found online at http://dx.doi.org/10.7717/peerj-cs.1904#supplemental-information.

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
