# Peer review of "Text-image semantic relevance identification for aspect-based multimodal sentiment analysis"

_PeerJ Computer Science, doi:10.7717/peerj-cs.1904_

## Round 0.1 · original submission · Major Revisions

Dear authors,

Thank you for submitting your article. Reviewers have now commented on your article and suggested major revisions. When submitting the revised version of your article, it will be better to address the following:

1- The research gaps and contributions should be clearly summarized in the introduction section. Please evaluate how your study is different from others in the related section.

2- The values for the parameters of the algorithms selected for comparison are not given.

3- The paper lacks the running environment, including software and hardware. The analysis and configurations of experiments should be presented in detail for reproducibility. It is convenient for other researchers to redo your experiments and this makes your work easy acceptance. A table with parameter settings for experimental results and analysis should be included in order to clearly describe them.

4- Clarifying the study’s limitations allows the readers to better understand under which conditions the results should be interpreted. A clear description of limitations of a study also shows that the researcher has a holistic understanding of his/her study. However, the authors fail to demonstrate this in their paper. The authors should clarify the pros and cons of the methods. What are the limitation(s) methodology(ies) adopted in this work? Please indicate practical advantages, and discuss research limitations.

5- Explanation of the equations should be checked. All variables should be written in italics as in the equations. Equations should be used with correct equation numbers within the text.

6- Please include future research directions.

7- Graphics and charts need more explanation.

8- The conclusion section is indicative, but it might be strengthened to highlight the importance and applicability of the work done with some more in-depth considerations, to summarize the findings, and to give readers a point of reference. Additional comments about the reached results should be included.

9- Reviewer 1 has requested that you cite specific references. You are welcome to add it/them if you believe they are relevant. However, you are not required to include these citations, and if you do not include them, this will not influence my decision.

**Language Note:** The review process has identified that the English language must be improved. PeerJ can provide language editing services - please contact us at copyediting@peerj.com for pricing (be sure to provide your manuscript number and title). Alternatively, you should make your own arrangements to improve the language quality and provide details in your response letter. – PeerJ Staff

·

Basic reporting

no comment

Experimental design

no comment

Validity of the findings

no comment

Additional comments

This paper proposes a novel model called Text-Image Semantic Relevance Identification (TISRI) to address the task of Aspect-Based Multimodal Sentiment Analysis (ABMSA). TISRI introduces a module to calculate semantic similarity between text and image. TISRI uses different networks to extract linguistic and visual features, and then fuses the representations. It constructs an "image gate" to dynamically control image information based on relevance to the text. An Image Feature Auxiliary Reconstruction layer is also included to refine image features using adjective-noun pairs extracted from images. Furthermore, TISRI employs attention to generate the final image representation by focusing on text-aligned regions. Extensive experiments on two Twitter datasets demonstrate that TISRI outperforms other state-of-the-art unimodal and multimodal baselines. Ablation studies validate the importance of key components like the relevance identification, feature reconstruction layer, and image gate. However, here are some points:

1.The manuscript would benefit from a more comprehensive literature review to properly situate the presented work within the context of prior advancements in the field. A more robust discussion of related work is needed.

2.The aspect selection and preprocessing steps could be described in greater algorithmic detail to facilitate replication. More information is needed regarding the identification and treatment of target terms.

3.The neural network architectures would benefit from additional technical specification and validation. Explicit justification for design choices and ablation studies are important to fully evaluate model configurations.

4.Potential limitations of the twitter corpora for evaluating real-world generalizability merit discussion. Alternative or additional datasets could bolster ecological validity.

5.The discussion and conclusions overstate contributions and understate outstanding questions. More modesty and identification of future research directions would improve manuscript balance.

6. The cited literature is not new enough. For example, you can refer to some new integrated learning methods:
(1)Yang, L., Wang, J., Na, J. C., & Yu, J. (2023). Generating paraphrase sentences for multimodal entity-category-sentiment triple extraction. Knowledge-Based Systems, 278, 110823.
(2)Xiao, L., Wu, X., Yang, S., Xu, J., Zhou, J., & He, L. (2023). Cross-modal fine-grained alignment and fusion network for multimodal aspect-based sentiment analysis. Information Processing & Management, 60(6), 103508.
(3)Yu, J., Chen, K., & Xia, R. (2022). Hierarchical interactive multimodal transformer for aspect-based multimodal sentiment analysis. IEEE Transactions on Affective Computing.

7. English writing can be improved to make the paper friendlier to the readers.

·

Basic reporting

Regarding the basic reporting of this manuscript, overall, the presentation and structure appears sound. The introduction provides a clear overview of the field and motivation for this work, adequately situating it within the relevant literature. Technical terms are well-defined throughout, and the problem being addressed is articulated with suitable background context.

The methodology section appropriately outlines each component of the proposed model at a technical level. Figures and tables are relevant, well-labelled and serve to effectively illustrate key aspects of the approach. Raw data has also been made available, satisfying basic reporting standards.

The results discuss evaluation of the method against appropriate baselines on established benchmarks. Performance claims are supported by quantitative results presented across consistent metrics. An ablation study further validates contribution of individual elements.

Several proof-reading passes may serve to strengthen the language throughout. Minor editorial polishing could also improve flow and readability in places. Some extension of discussions addressing limitations and future directions would further contextualize findings.

Experimental design

While the overall experimental design presented appears sufficiently rigorous, a few aspects could potentially be strengthened:

Error or failure cases are discussed but potential causal factors behind misclassifications are not always deeply explored. Further probing unsuccessful predictions could offer insight into limitations and avenues for improvement.

While datasets are established, sample characteristics and any imbalances are not fully detailed. More comprehensive reporting of data properties could contextualize results.

Although novelty is cursorily claimed, precisely how findings advance the state-of-the-art or differ from prior work is not entirely clear. Direct comparisons against the most related and successful prior efforts would strengthen impact assessment.

Validity of the findings

Regarding the validity of findings presented by this work, overall the quantitative evaluation and analysis appears largely rigorous. However, a few aspects could potentially be expanded upon:

While performance metrics are clearly reported across datasets and baselines, details on their calculations and any nuances in implementation are lacking. Clarifying such technical specifics would strengthen claims.

Potential overfitting concerns are not addressed, such as through monitoring validation performance during training or testing additional regularization. Demonstrating generalizability could further bolster confidence in results.

Additional comments

While the overall work appears technically sound, a few minor issues could potentially be addressed:

First, the definition of ABMSA described in Abstract seems to be wrong.

Second, related efforts are discussed but high-level similarities and differences versus the most directly comparable prior works are not explicitly drawn out.

Third, the conclusions characterize accomplishments yet do not clearly delineate specific technical or conceptual advances offered over prior methods.

---

## Round 0.2 · accepted · Accept

Dear authors,

Thank you for the revision and for clearly addressing all the reviewers' comments. I confirm that the paper is improved and addresses the concerns of the reviewers. Your paper is now acceptable for publication in light of this revision.

Best wishes,

·

Basic reporting

no comment

Experimental design

no comment

Validity of the findings

no comment

Additional comments

Authors have addressed all the comments and updated the manuscript accordingly.

·

Basic reporting

Accept

Experimental design

Accept

Validity of the findings

Accept

Additional comments

Accept